# Difficulties in Managing Children’s Learning among Caregivers of Children with Attention-Deficit/Hyperactivity Disorder during the COVID-19 Pandemic in Taiwan: Association with Worsened Behavioral and Emotional Symptoms

**DOI:** 10.3390/ijerph192113722

**Published:** 2022-10-22

**Authors:** Chao-Ying Chen, Jung-Sheng Chen, Chung-Ying Lin, Ray C. Hsiao, Ching-Shu Tsai, Cheng-Fang Yen

**Affiliations:** 1School of Physical Therapy and Graduate Institute of Rehabilitation Science, College of Medicine, Chang Gung University, Taoyuan 33302, Taiwan; 2Department of Internal Medicine/Pediatric, New Taipei City Tucheng Hospital (Chang Gung Medical Foundation), New Taipei City 23652, Taiwan; 3Department of Medical Research, E-Da Hospital, Kaohsiung 82445, Taiwan; 4Institute of Allied Health Sciences, College of Medicine, National Cheng Kung University, Tainan 70101, Taiwan; 5Biostatistics Consulting Center, National Cheng Kung University Hospital, College of Medicine, National Cheng Kung University, Tainan 70101, Taiwan; 6Department of Public Health, College of Medicine, National Cheng Kung University, Tainan 70101, Taiwan; 7Department of Occupational Therapy, College of Medicine, National Cheng Kung University, Tainan 70101, Taiwan; 8Department of Psychiatry and Behavioral Sciences, University of Washington School of Medicine, Seattle, WA 98295, USA; 9Department of Psychiatry, Seattle Children’s, Seattle, WA 98105, USA; 10Department of Child and Adolescent Psychiatry, Chang Gung Memorial Hospital, Kaohsiung Medical Center, Kaohsiung 83301, Taiwan; 11School of Medicine, Chang Gung University, Taoyuan 33302, Taiwan; 12Department of Psychiatry, Kaohsiung Medical University Hospital, Kaohsiung 80756, Taiwan; 13Department of Psychiatry, School of Medicine, Kaohsiung Medical University, Kaohsiung 80708, Taiwan; 14College of Professional Studies, National Pingtung University of Science and Technology, Pingtung 91201, Taiwan

**Keywords:** attention-deficit/hyperactivity disorder, COVID-19, learning, caregiver, mental health

## Abstract

School closures during the COVID-19 pandemic have interfered with children’s learning. The aim of this study was to investigate the difficulties in managing children’s learning at home and attending afterschool learning programs and their related factors among caregivers of children with attention-deficit/hyperactivity disorder (ADHD) during the COVID-19 pandemic. In total, 252 caregivers of children with ADHD completed a questionnaire collecting difficulties in managing children’s learning, parenting styles, children’s worsened symptoms of ADHD, oppositional defiant disorder (ODD) and emotion, and increased Internet use. Multivariate logistic regression models were used to examine the factors related to caregiver difficulties in managing children’s learning and children’s worsened ADHD, ODD, and emotional symptoms. In total, 85.3% of the caregivers had difficulty in asking their children to learn at home; 28.2% had difficulty in taking children to afterschool learning programs. Children’s worsened anger was significantly associated with higher caregiver difficulty in asking children to learn at home, whereas parental overprotection was significantly associated with lower caregiver difficulty in asking children to learn at home. Worsened hyperactivity and opposition were significantly associated with higher caregiver difficulty in taking children to attend afterschool learning programs. Interventions for enhancing caregivers’ skills to manage children’s learning and children’s behavioral and emotional symptoms should take the related factors found in this study into consideration.

## 1. Introduction

The COVID-19 pandemic has considerably affected the daily routines [1,2,3,4], family life [3,5], and social interactions [6] of children. School closures during the COVID-19 pandemic have interfered with children’s learning [4,6,7]. Studies have reported that one-fifth to one-third of children have encountered difficulties in remote learning at home [4,6] and that difficulties in learning considerably increased the risk of low health-related quality of life among children during COVID-19 lockdowns [8]. A high proportion of children have also experienced considerable worsening of emotional symptoms, conduct problems, hyperactivity, and inattention during COVID-19 lockdowns [3,4,5,9,10]. Changes in learning conditions have been reported to be associated with children’s behavioral and emotional problems. Compared with children who attended in-person schooling, children participating in remote learning have experienced more mental health difficulties during the COVID-19 pandemic [2,11]. Both cross-sectional [3,9,12,13] and longitudinal studies [5] have demonstrated a significant association between difficulties in learning and emotional and behavioral problems in children.

The core symptoms of attention-deficit/hyperactivity disorder (ADHD) including inattention, hyperactivity, and impulsivity increased the difficulties in learning among children with ADHD (CADHD) before the COVID-19 pandemic; CADHD encountered more academic problems such as lower grades, grade repetition, and increased school drop-out compared with those without ADHD [14,15,16,17]. Research has found that CADHD experienced more difficulties in remote learning compared with those without ADHD during the COVID-19 pandemic [18]; in particular, CADHD exhibited shorter attention spans, less spontaneous commitment, and less autonomy in remote learning during the COVID-19 pandemic [19]. Studies have confirmed that not only ADHD symptoms but also emotional dysregulation and oppositional defiant disorder (ODD) have negative influences on CADHD’s learning [20,21]. Both emotional dysregulation and ODD were commonly presented in children with ADHD before the COVID-19 pandemic [22,23,24]. Children with ADHD were more likely than children without ADHD to experience an increase in inattentive, hyperactive/impulsive, oppositional/defiant, and emotional symptoms during the COVID-19 pandemic [25]. Higher negative affect and greater concentration difficulties because of COVID-19 were significantly associated with greater difficulties in remote learning in adolescents with ADHD, but no such associations were reported in those without ADHD [18]. Emotional problems [26,27], inattention [26,27], hyperactivity [26], and ODD symptoms [26] have been associated with difficulties in remote learning among CADHD during the COVID-19 pandemic [27].

Several problems relating to learning difficulties among CADHD during the COVID-19 pandemic warrant examination. First, caregivers have a key role in providing support for their children’s home learning during the COVID-19 pandemic. However, research has reported that, compared with caregivers of children without ADHD, caregivers of CADHD had a lower level of confidence in managing remote learning and a higher level of difficulties in supporting home learning and home–school communication during the COVID-19 pandemic [18]. Research has also reported that families of CADHD who experienced difficulties in remote learning often had a negative emotional climate and low parenting efficacy [12,26]. The relationship between worsened ADHD, ODD, and emotional symptoms and caregiver difficulties in managing children’s learning during the COVID-19 pandemic warrants examination. Second, research before the COVID-19 pandemic found that CADHD and their family members may encounter difficulties in supporting, interacting, and communicating with each other [28]. The interactions between caregivers and their CADHD before the COVID-19 pandemic may influence caregivers’ difficulties in managing their children’s behaviors during the pandemic. Parenting styles indicate caregivers’ long-term attitudes and behaviors toward their children that commence in the early developmental stages [29]. The associations between parenting styles and caregiver difficulties in managing the learning of their CADHD during the COVID-19 pandemic have not been researched. Third, the spectrum of learning outside school is not limited to learning at home such as remote learning and homework; attendance of afterschool learning programs for schoolwork, extracurricular activities, or sports also plays a key role in child education. The associations of caregiver difficulty level in taking their CADHD to afterschool learning programs during the COVID-19 pandemic with worsened ADHD, ODD, and emotional symptoms and with various parenting styles warrant further study. Fourth, CADHD exhibited greater digital media use than did those without ADHD during the COVID-19 pandemic [30]. Research has also reported that CADHD who reported high levels of difficulty in remote learning spent considerably more time playing video games and visiting social networking sites compared with CADHD who reported low levels of difficulty in remote learning [26]. However, the associations between increased Internet use with caregiver difficulties in managing the home learning of their CADHD and in facilitating attendance at afterschool learning programs have not been investigated.

Taiwan experienced a severe outbreak of COVID-19 between May and July of 2021, at which time schools were closed for the first time since the start of the COVID-19 pandemic. This study investigated the child-related factors (worsened ADHD, ODD, and emotional symptoms and increased Internet use) and caregiver parental styles related to caregiver difficulties in managing the learning of their CADHD (learning at home and attending afterschool learning programs) during the COVID-19 pandemic. We hypothesized that children’s worsened ADHD, ODD, and emotional symptoms and increased Internet use would be significantly associated with caregiver difficulties in managing the learning of their CADHD. We also hypothesized that various parenting styles would exhibit different associations with caregiver difficulties in managing the learning of their CADHD.

## 2. Methods

### 2.1. Participants

Caregivers of CADHD who were aged 6–18 years were recruited at the child and adolescent psychiatric outpatient clinics of two hospitals in Kaohsiung, Taiwan, during the period of August 2021 to January 2022. Formal medical diagnoses of ADHD were determined according to the *Diagnostic and Statistical Manual of Mental Disorders, Fifth Edition* [31]. Caregivers were excluded from this study if they had any cognitive impairments that prevented them from understanding the purpose and experimental procedures of the present study. The study was approved by the Institutional Review Board of Chang Gung Medical Foundation (202002118B0C501) and Kaohsiung Medical University Hospital (KMUHIRB-E(I)-20200408), and all assessments were conducted after receiving informed consent. In total, 252 caregivers of CADHD participated in this study and returned self-reported questionnaires.

### 2.2. Measures

All measures were completed by parents of CADHD. The following describe the information that was collected.

#### 2.2.1. Caregiver Difficulties in Managing Children’ Learning

Two items were designed to assess the difficulties caregivers encountered managing child learning during the COVID-19 pandemic. One item asked whether caregivers found it difficult to ask their children to learn at home, including doing remote learning and completing homework (*yes* or *no*) [32]; another asked whether caregivers had difficulties taking their children to attend afterschool learning programs for schoolwork, extracurricular activities, or sports (*yes* or *no*). A previous study in 2020 using the same item found that caregiver difficulties in managing children’ learning at home was significantly associated with caregivers’ poor mental health state in the COVID-19 pandemic [32].

#### 2.2.2. Changes in ADHD, ODD, and Emotional Symptoms and Internet Use

Nine items were designed to assess changes in ADHD symptoms (inattention, hyperactivity, and impulsivity), ODD symptoms (anger, opposition, and revenge), emotional symptoms (depression and anxiety), and Internet use during the COVID-19 pandemic among the children [32]. All responses were dichotomous: same as before COVID-19 or worse than before COVID-19. A previous study in 2020 using the same items found that children’s worsened ADHD, ODD, and emotional symptoms were significantly associated with caregivers’ poor mental health state in the COVID-19 pandemic [32].

#### 2.2.3. Parenting Styles

The Chinese version of the Parental Bonding Instrument (PBI), Parent Version was used to assess the parenting styles and how caregivers bonded with their children [28,33,34]. This 25-item instrument was originally developed as a three-factor model to evaluate parental affection and care (12 items), overprotection (7 items), and authoritarianism (6 items). Each testing item is scored using a 4-point Likert scale that ranges from 1 (*strongly agree*) to 4 (*strongly disagree)*; 12 items need to be reverse coded. Scores are then summed for the final interpretation. A higher score for parental affection and care indicates a more affectionate and caring parenting style, a higher score for authoritarian parenting indicates a parenting style that is more encouraging of behavioral freedom, and a higher score for parental overprotection indicates a parenting style more likely to deny psychological autonomy to a child. The internal consistency, validity, and test–retest reliability of the PBI were reported to be good in previous studies [28,33,34].

#### 2.2.4. ADHD and ODD Symptoms

The traditional Chinese version of the Swanson, Nolan, and Pelham Rating Scale-IV (SNAP-IV)-Parent Form, was used to assess the severity of ADHD and ODD symptoms for children in the present study [35,36]. The SNAP-IV is a reliable and valid instrument composed of three subsets including inattention, hyperactivity and impulsivity, and opposition and defiance subsets, for a total of 26 items. The severity of each testing item is scored using a 4-point Likert scale ranging from 0 (*not at all)* to 3 (*extremely*). The sum scores of each subset were used for further statistical analyses.

#### 2.2.5. Demographics

Children’s age, sex, comorbidity diagnoses, and frequency of receiving medication treatment were assessed using the background information sheet. Specifically, age was assessed using year as the unit, sex was assessed using a dichotomous answer (boy or girl), and frequency of receiving medication treatment was assessed using a 4-point Likert scale (*never*, *rarely*, *sometimes*, and *often*). Frequency of receiving medication treatment was then converted into a dichotomous scale to indicate whether the child received regular treatment (i.e., *never* to *sometimes* as no; *often* as yes); the results thereof were used in subsequent logistic regression analyses.

### 2.3. Data Analysis

Frequency with percentage and mean with standard deviation (SD) were used to summarize the children’s demographic characteristics, clinical characteristics, difficulties in learning, parental bonding, and changes in ADHD, ODD, and emotional symptoms and Internet use. Multivariate logistic regression analysis was used to examine the factors related to caregiver difficulties in managing child learning. Changes in ADHD (attention, hyperactivity, impulsivity), ODD (anger, opposition, revenge) and emotional symptoms (depression, anxiety), and change in Internet use and parenting styles (parental affection and care, overprotection, and authoritarianism) were entered for selection using the forward stepwise method. Sex (reference group: girl), age, regular treatment (reference group: no), and ADHD and ODD symptoms as measured using SNAP-IV were entered as control variables. In order to compare caregiver difficulties in managing CADHD’s learning between caregivers of children with and without severe worsened ADHD symptoms, we summed up the scores of worsened inattention, hyperactivity, and impulsivity and classified CADHD with the top 25% and bottom 25% of worsened total ADHD symptoms as CADHD with and without severe worsened ADHD symptoms, respectively. Similarly, we summed up the scores of worsened anger, opposition, and revenge and classified CADHD with the top 25% and bottom 25% of worsened total ODD symptoms as CADHD with and without severe worsened ODD symptoms, respectively. The associations of severe worsened ADHD and ODD symptoms with caregiver difficulties in managing their CADHD’s learning were examined using multivariate logistic regression analysis. Adjusted odds ratios (*AOR*) and 95% confidence intervals (CI) were calculated for all logistic regression models. SPSS version 20.0 (IBM, Armonk, NY, USA) was used to perform all the statistical analyses.

## 3. Results

Table 1 additionally reports the children’s demographics, treatment, ADHD and ODD symptoms, parenting styles information, and caregivers’ difficulties in managing child’s learning. The children had a mean age of 9.61 (SD = 2.39) years. Approximately four-fifths of the children (*n* = 200; 79.4%) were boys, and the majority of them received regular medical treatment (*n* = 212; 84.1%). Children’s mean (SD) scores for the dimensions of inattention, hyperactivity/impulsivity, and ODD symptoms were 12.88 (5.83), 9.93 (6.17), and 9.33 (5.92), respectively. The mean (SD) PBI scores for the affection care, overprotection, and authoritarian parenting dimensions were 37.08 (5.16), 13.75 (3.32), and 12.24 (2.67), respectively. Over four-fifths of the caregivers had difficulty in asking their children to learn at home (*n* = 215; 85.3%); more than one-quarter had difficulty in taking their children to afterschool learning programs (*n* = 71; 28.2%).

The top three worsened ADHD and ODD symptoms among CADHD during the COVID-19 pandemic reported by the caregivers were opposition (*n* = 122; 48.4%), anger (*n* = 113; 44.8%), and impulsivity (*n* = 102; 40.5%). Over one-fifth of the children had worsened emotional symptoms, including depression (*n* = 57; 22.6%) and anxiety (*n* = 59; 23.4%). Moreover, over two-thirds of the children demonstrated increased Internet use (*n* = 169; 67.1%; see Table 2).

The results of multivariate logistic regression models are shown in Table 3. Worsened anger (*AOR* = 4.36; 95% CI = 1.56, 12.18) was significantly associated with higher caregiver difficulty in asking children to learn at home, whereas parental overprotection was significantly associated with lower caregiver difficulties in asking children to learn at home (*AOR* = 0.88; 95% CI = 0.78, 0.99). Worsened attention, hyperactivity, impulsivity, opposition, revenge, depression, anxiety and Internet use, and parenting styles of affection/care and authoritarianism were not significantly associated with caregiver difficulties in asking children learning at home.

Worsened hyperactivity (*AOR* = 3.46; 95% CI = 1.73, 6.91) and opposition (*AOR* = 1.99; 95% CI = 1.01, 3.95) were significantly associated with higher caregiver difficulty in taking children to attend afterschool learning programs. Worsened attention, impulsivity, anger, revenge, depression, anxiety and Internet use, and parenting styles were not significantly associated with caregiver difficulties in taking children to attend afterschool learning programs.

We identified 60 and 51 CADHD with and without severe worsened ADHD symptoms, respectively, using the cutoffs described in Section 2; we also identified 55 and 47 CADHD with and without severe worsened ODD symptoms, respectively. The associations of severe worsened ADHD and ODD symptoms with caregiver difficulties in managing their CADHD’s learning were further examined using forward multivariate logistic regression analysis by controlling for the covariates. The results indicated that severe worsened ADHD symptoms were significantly associated with caregiver difficulties in managing CADHD learning at home (*AOR* = 1.60; 95% CI = 1.30, 1.97), whereas severe worsened ODD symptoms were significantly associated with caregiver difficulties in taking children to after school learning program (*AOR* = 1.35; 95% CI = 1.17, 1.55).

## 4. Discussion

### 4.1. Main Findings

The present study found that over four-fifths of caregivers had difficulties in managing the learning of their CADHD at home, including doing remote learning and completing schoolwork. The results indicated that when faced with school closures, learning at home proved a challenge for caregivers and their CADHD. The learning challenges were not limited to home learning but also to children’s attendance of afterschool learning programs. Taiwan has never imposed a lockdown because of the COVID-19 pandemic; therefore, children could continue their afterschool learning programs for schoolwork, extracurricular activities, or sports even during the period of school closures. However, over one-quarter of caregivers reported difficulties taking their children to attend these afterschool learning programs during the pandemic. The results of this study indicated that the COVID-19 pandemic might have had a considerable influence on the learning behaviors of CADHD. The after-effects of learning interference from the COVID-19 pandemic among CADHD warrant investigation through follow-up studies; remedial programs may be necessary if the ill effects of home learning are severe and result in persistent problems with learning among CADHD.

The present study found that worsened ADHD, ODD, and emotional symptoms during the COVID-19 pandemic were prevalent among CADHD. The results of the present study are congruent with those of other studies on CADHD during the COVID-19 pandemic [37,38,39]. Changes in their routine lives, limited range of life, decreased peer interaction, increased family conflicts, and worries about contracting COVID-19 during the pandemic may have worsened ADHD, ODD, and emotional symptoms among CADHD. The present study also determined that children’s worsened anger was significantly associated with higher caregiver difficulty in asking children to learn at home, and children’s worsened hyperactivity and opposition were significantly associated with higher caregiver difficulty in taking children to attend afterschool learning programs. Although the cross-sectional study design precludes any possibility of determining the temporal relationship between caregiver difficulties in managing the learning of their CADHD and worsening psychopathologies, the two variables may reciprocally influence each other. CADHD who have worsened ODD symptoms such as anger and opposition might have low motivation to cooperate with caregivers by participating in remote learning, completing homework, or attending afterschool learning programs. Alternatively, the conflicts regarding home learning between caregivers and their CADHD may exacerbate children’s anger and opposition.

The present study also determined that caregivers with higher parental overprotection reported lower difficulty in asking children to learn at home. Caregivers adopting a parenting style of overprotection may deny their children’s psychological autonomy. Children with low psychological autonomy may passively cooperate with caregiver management. Although caregivers with the parental overprotection style might feel less difficulty in asking their children to participate in home learning during the COVID-19 pandemic, CADHD with low psychological autonomy may lack the spontaneous motivation and planning necessary for study. The long-term influence of parenting overprotection on the learning of CADHD warrants further study.

A meta-analysis study demonstrated that no gender difference in academic performance existed in CADHD [40]. ADHD impaired CADHD’s academic performance since the early stage of school career [41]. The present study found that gender and age were not significantly associated with caregiver difficulties in managing their child’s learning. However, the present study only examined caregiver difficulties in managing their CADHD’s learning during the COVID-19 pandemic but did not allow for understanding academic performance problems at school. Research has found that inattention but not hyperactivity/impulsivity or ODD symptoms predicted persistent patterns of academic problems in CADHD [20,42]. Moreover, academic performance outcomes were worse in individuals with untreated ADHD compared with non-ADHD controls [43,44]. The present study found that regular treatment for ADHD, inattention, hyperactivity, impulsivity/impulsivity and ODD symptoms were not significantly associated with caregiver difficulties in managing their child’s learning. In addition to the difference in research topics between this study (caregiver difficulties in managing their child’s learning) and previous studies (academic performance problems at school), the special learning situations during the COVID-19 pandemic such as school closures and remote learning may contribute to the inconsistency of the study results.

### 4.2. Practical Implication and Future Directions

Based on the results of the present study, we suggest that intervention programs for difficulties in managing child learning among caregivers of CADHD should involve multiple elements. First, caregiver difficulties in managing child learning should be routinely assessed for early detection. Enhancing caregivers’ managing skills to support children’s learning motivation and ability during the COVID-19 pandemic may help to reduce the difficulties of caregivers. Second, children’s ADHD and ODD symptoms should also be routinely assessed and intervened. Enhancing caregivers’ skills to increase children’s cooperation with caregivers to maintain their daily routine, control their impulsivity and anger, and regulate their Internet use during the COVID-19 pandemic may reduce the severity of behavioral and psychological symptoms. Third, the continuation of effective pharmacological treatment for children’s ADHD is recommended [45]. The implementation of behavioral interventions is conducive to maintaining the psychological well-being of CADHD during the COVID-19 pandemic [45]. Fourth, medical and educational professionals may deliver medical and educational support online for caregivers and CADHD during lockdowns or other difficult situations during or following the COVID-19 pandemic [46].

### 4.3. Limitations

This study has several limitations. First, the study collected data from the caregivers who visited child psychiatric outpatient clinics to treat their CADHD; thus, whether the results of this study can be generalized to caregivers of CADHD who do not search for medical help for treating their CADHD warrants examination. Second, this cross-sectional study could not determine the temporal relationships of children’s worsened ADHD, ODD, and emotional symptoms and increased Internet use with caregivers’ difficulties in managing their children’s learning. Third, this study collected all the data from caregivers; the possible problem of shared-method variance could not be ruled out. Moreover, although the previous study using the same research questions found that caregivers’ difficulty in managing child’s learning at home and children’s worsened ADHD, ODD, and emotional symptoms were significantly associated with caregivers’ poor mental health state in the COVID-19 pandemic [32], the validity and reliability of the questions assessing caregivers’ difficulties in managing child’s learning and children’s worsened ADHD, ODD, and emotional symptoms warrant further study. Fourth, this study did not enroll the caregivers of children without ADHD for comparison. We therefore could not state that children without ADHD did not experience the same negative repercussions and that their caregivers did not suffer similar difficulties in managing child’s learning during the COVID-19 pandemic.

## 5. Conclusions

The present study found that a high proportion of caregivers had difficulties in managing the learning of their CADHD at home and attending afterschool learning programs during the COVID-19 pandemic. Because caregivers have a key role in providing support for their children’s home learning during the COVID-19 pandemic, enhancing caregivers’ managing skills to support children’s learning motivation and ability during the COVID-19 pandemic is necessary to reduce the difficulties of caregivers. The after-effects of learning interference from the COVID-19 pandemic among children with ADHD warrant investigation through follow-up studies. Children’s worsened anger, hyperactivity, and opposition were significantly associated with higher caregiver difficulty in managing their CADHD’s learning; therefore, children’s ADHD and ODD symptoms should also be routinely assessed and intervened. Caregivers with higher parental overprotection reported lower difficulty in asking children to learn at home; however, the long-term influence of parenting overprotection on the learning of CADHD warrants further study. Behavioral and medication interventions should be conducted to maintain the psychological well-being of CADHD during the COVID-19 pandemic.

## Figures and Tables

**Table 1 ijerph-19-13722-t001:** Characteristics of the participants and caregivers’ difficulties in managing ADHD children’s learning (N = 252).

	*n* (%)	M (SD)
Age (years)		9.61 (2.39)
Sex		
Boy	200 (79.4)	
Girl	52 (20.6)	
Medicine treatment for ADHD		
Never		21 (8.3)
Rarely		7 (2.8)
Sometimes		12 (4.8)
Often		212 (84.1)
Inattention symptom ^a^		12.88 (5.83)
Hyperactivity/impulsivity symptom ^a^		9.93 (6.17)
ODD symptom ^a^		9.33 (5.92)
Parental affection care ^b^		37.08 (5.16)
Parental overprotection ^b^		13.75 (3.32)
Parental authoritarianism ^b^		12.24 (2.67)
Caregivers’ difficulty in asking children learning at home		
Yes	215 (85.3)	
No	37 (14.7)	
Caregivers’ difficulty in taking children to attend after school learning programs		
Yes	71 (28.2)	
No	181 (71.8)	

^a^ Assessed using Swanson, Nolan, and Pelham Rating Scale-IV; ^b^ Assessed using Parental Bonding Instrument; ADHD = attention deficit/hyperactivity disorder; ODD = oppositional defiant disorder.

**Table 2 ijerph-19-13722-t002:** Changes in ADHD, ODD, and emotional symptoms and internet use of children with ADHD during the COVID-19 pandemic (N = 252).

	*n* (%)	
	Same as or Better before the COVID-19 Pandemic	Worse Than before the COVID-19 Pandemic
Changes in attention	154 (61.1)	98 (38.9)
Changes in hyperactivity	172 (68.3)	80 (31.7)
Changes in impulsivity	150 (59.5)	102 (40.5)
Changes in anger	139 (55.2)	113 (44.8)
Changes in opposition	130 (51.6)	122 (48.4)
Changes in revenge	216 (85.7)	36 (14.3)
Changes in depression	195 (77.4)	57 (22.6)
Changes in anxiety	193 (76.6)	59 (23.4)
Changes in internet use	83 (32.9)	169 (67.1)

ADHD = attention deficit/hyperactivity disorder; ODD = oppositional defiant disorder.

**Table 3 ijerph-19-13722-t003:** Factors explaining caregivers’ difficulties in managing ADHD children’s learning in the binary logistic regression model.

	*AOR* (95% CI)/*p*-Value	
	Difficulty in Asking Children Learning at Home	Difficulty in Taking Children to after School Learning Programs
*Step 1*		
Sex (Ref: girl)	2.13 (0.92, 4.94)/0.08	0.75 (0.36, 1.54)/0.43
Age	1.05 (0.89, 1.24)/0.55	1.03 (0.90, 1.18)/0.70
Regular treatment (Ref: no)	1.29 (0.89, 1.88)/0.18	0.89 (0.65, 1.23)/0.89
Inattention symptom	1.06 (0.97, 1.15)/0.23	1.04 (0.97, 1.11)/0.31
Hyperactivity/impulsivity symptom	1.01 (0.92, 1.11)/0.83	1.04 (0.97, 1.12)/0.23
ODD symptom	1.02 (0.91, 1.13)/0.79	0.94 (0.88, 1.01)/0.09
*Step 2*		
Worsened anger	4.36 (1.56, 12.18)/0.005	--
Worsened hyperactivity	--	3.46 (1.73, 6.91)/<0.001
Worsened opposition	--	1.99 (1.01, 3.95)/0.048
Parental overprotection	0.88 (0.78, 0.99)/0.03	--
Nagelkerke R^2^	0.206	0.160

Note. All variables in Step 1 were entered in the logistic regression model; variables in Step 2 were selected in the logistic regression model using forward stepwise method. In Step 2, worsened ADHD (attention, hyperactivity, impulsivity) and ODD symptoms (anger, opposition, revenge), worsened emotional symptoms (depression, anxiety), worsened internet use, parental affection care, parental overprotection, and parental authoritarianism were entered for selection. ADHD = attention deficit/hyperactivity disorder; *AOR* = adjusted odds ratio; CI = confidence interval; ODD = oppositional defiant disorder.

## Data Availability

The data used in this study are available upon reasonable request to the corresponding authors.

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
