# Peer review of "Difficulties in Managing Children’s Learning among Caregivers of Children with Attention-Deficit/Hyperactivity Disorder during the COVID-19 Pandemic in Taiwan: Association with Worsened Behavioral and Emotional Symptoms"

_ijerph, 2022, doi:10.3390/ijerph192113722_

Round 1

Reviewer 1 Report

The manuscript examines caregiver difficulties in managing at-home learning and afterschool program attendance during COVID among children with ADHD. There is still a paucity of research on the effects of COVID on youth and caregiver difficulties, especially among youth with behavioral disorders. The following are offered as points of clarity to further strengthen the submission:

1.       The manuscript title implies that CADHD is the focal group, yet just as much attention and analysis involved ODD. The authors may wish to explain the distinction and why ODD symptoms figured so prominently throughout the research. Of note, I do not disagree with it at all, just believe it warrants some explanation.

2.       The limitations section does mention in a single sentence that there was no non-ADHD comparison group. The repercussions of this should be included as well (e.g., something such as “we therefore cannot state that youth without CAHDHD did not also experience may of the same negative repercussions and to the same extent as CADHD and that their caregivers did not also suffer similar difficulties”). Without a comparison group we do not know whether findings are generalizable to child in general or whether there is some unique aspects for those with CADHD.

3.       Similarly, the data would allow for examining whether the extent of ADHD and/or ODD symptoms was related to caregiver difficulties (perhaps selecting the top 25% and bottom 25% in terms of symptoms and see if relationships to caregiver difficulties are similar/different).

4.       The authors should note and discuss the null findings in relation to no sex, age, regular treatment, hyperactivity/impulsivity, and ODD symptoms. Were all of those relationships of no difference expected, or were caregiver difficulties expected worse for those with boys, etc.?

5.       The discussion and conclusion both mention monitoring Internet use, yet Internet does not appear to be significant in the multivariate models. The authors may wish to elaborate on why they still believe Internet use to be a potential problem.

6.       Lastly, the data presented does not allow for understanding whether any of the measures were related to actually performing worse at school (lower grades, etc.), only that parents expressed more difficulties during COVID. That is an important distinction, I think.

Thank you for this important work!

Author Response

We appreciated your valuable comments. As discussed below, we have revised our manuscript with underlines based on your suggestions. Please let us know if we need to provide anything else regarding this revision.

Comment

  1. The manuscript title implies that CADHD is the focal group, yet just as much attention and analysis involved ODD. The authors may wish to explain the distinction and why ODD symptoms figured so prominently throughout the research. Of note, I do not disagree with it at all, just believe it warrants some explanation.

Response

Thank you for your comment. We added the explanations for considering ODD symptoms in this study as below.

Studies have confirmed that not only ADHD symptoms but also emotional dysregulation and oppositional defiant disorder (ODD) have negative influences on CADHD’s learning [20,21]. Both emotional dysregulation and ODD were commonly presented in children with ADHD before the COVID-19 pandemic [22–24].” Please refer to line 81-84.

…ODD symptoms [26] have been associated with difficulties in remote learning among CADHD during the COVID-19 pandemic [27].” Please refer to line 91-92.

Comment

  1. The limitations section does mention in a single sentence that there was no non-ADHD comparison group. The repercussions of this should be included as well (e.g., something such as “we therefore cannot state that youth without CAHDHD did not also experience may of the same negative repercussions and to the same extent as CADHD and that their caregivers did not also suffer similar difficulties”). Without a comparison group we do not know whether findings are generalizable to child in general or whether there is some unique aspects for those with CADHD.

Response

Thank you for your suggestion. We added this sentence into the Limitation section as below. Please refer to line 375-378

We therefore could not state that children without ADHD did not experience the same negative repercussions and that their caregivers did not suffer similar difficulties in managing child’s learning during the COVID-19 pandemic.

Comment

  1. Similarly, the data would allow for examining whether the extent of ADHD and/or ODD symptoms was related to caregiver difficulties (perhaps selecting the top 25% and bottom 25% in terms of symptoms and see if relationships to caregiver difficulties are similar/different).

Response

Thank you for your suggestion. We conducted the analysis and added the results as below into the revised manuscript.

In order to compare caregiver difficulties in managing CADHD’s learning between caregivers of children with and without severe worsened ADHD symptoms, we summed up the scores of worsened inattention, hyperactivity and impulsivity and classified CADHD with the top 25% and bottom 25% of worsened total ADHD symptoms as CADHD with and without severe worsened ADHD symptoms, respectively. Similarly, we summed up the scores of worsened anger, opposition and revenge and classified CADHD with the top 25% and bottom 25% of worsened total ODD symptoms as CADHD with and without severe worsened ODD symptoms, respectively. The associations of severe worsened ADHD and ODD symptoms with caregiver difficulties in managing their CADHD’s learning were examined using multivariate logistic regression analysis.” Please refer to line 210-220.

We identified 60 and 51 CADHD with and without severe worsened ADHD symptoms, respectively using the cutoffs described in Methods section; we also identified 55 and 47 CADHD with and without severe worsened ODD symptoms, respectively. The associations of severe worsened ADHD and ODD symptoms with caregiver difficulties in managing their CADHD’s learning were further examined using forward multivariate logistic regression analysis by controlling for the covariates. The results indicated that severe worsened ADHD symptoms were significantly associated with caregiver difficulties in managing CADHD learning at home (AOR = 1.60; 95% CI = 1.30, 1.97), whereas severe worsened ODD symptoms were significantly associated with caregiver difficulties in taking children to after school learning program (AOR = 1.35; 95% CI = 1.17, 1.55).” Please refer to line 272-281.

Comment

  1. The authors should note and discuss the null findings in relation to no sex, age, regular treatment, hyperactivity/impulsivity, and ODD symptoms. Were all of those relationships of no difference expected, or were caregiver difficulties expected worse for those with boys, etc.?

Response

Thank you for your comment. We added discussion regarding the roles of sex, age, regular treatment, hyperactivity/impulsivity, and ODD symptoms as below into the revised manuscript. Please refer to line 326-342.

A meta-analysis study demonstrated that no gender difference in academic performance existed in CADHD [40]. ADHD impaired CADHD’s academic performance since the early stage of school career [41]. The present study found that gender and age were not significantly associated with caregiver difficulties in managing their child’s learning. However, the present study only examined caregiver difficulties in managing their CADHD’s learning during the COVID-19 pandemic but not allowed for understanding academic performing problems at school. Research has found that inattention but not hyperactivity/impulsivity or ODD symptoms predicted persistent patterns of academic problems in CADHD [20,42]. Moreover, academic performance outcomes were worse in individuals with untreated ADHD compared with non-ADHD controls [43,44]. The present study found that regular treatment for ADHD, inattention, hyperactivity, impulsivity/impulsivity and ODD symptoms were not significantly associated with caregiver difficulties in managing their child’s learning. In addition to the difference in research topics between this study (caregiver difficulties in managing their child’s learning) and previous studies (academic performing problems at school), the special learning situations during the COVID-19 pandemic such as school close and remote learning may contribute the inconsistence of study results.

Comment

  1. The discussion and conclusion both mention monitoring Internet use, yet Internet does not appear to be significant in the multivariate models. The authors may wish to elaborate on why they still believe Internet use to be a potential problem.

Response

Thank you for your reminding. We deleted the descriptions regarding Internet use in the discussion and conclusion.

children’s ADHD and ODD symptoms should also be routinely assessed and intervened.” Please refer to line 345-346 and 389-390.

Comment

  1. Lastly, the data presented does not allow for understanding whether any of the measures were related to actually performing worse at school (lower grades, etc.), only that parents expressed more difficulties during COVID. That is an important distinction, I think.

Response

Thank you for your comment. We added it into the revised manuscript to remind the readers the distinction between academic performance and caregiver difficulties in managing child learning as below. Please refer to line 330-332.

“…the present study only examined caregiver difficulties in managing their CADHD’s learning during the COVID-19 pandemic but not allowed for understanding academic performing problems at school.

Reviewer 2 Report

The introduction is lax. The introduction does not provide any contextualisation that focuses or connects the difficulties that children with ADHD and their caregivers already have around behavioural symptoms and emotional symptoms. 

The methodology is well presented, simple but appropriate to the data. However, some of the measurement instruments are not pre-scored. At a scientific level, it does not stand out too much, it presents obvious hypotheses, but it is necessary to make the results known given the exceptional conditions of the pandemic.

The Participants section should note or name the presentation of the sample in more detail in Table 1. 

The instruments Caregiver Difficulties in Managing Children's Learning and Changes in ADHD, ODD, and Emotional Symptoms and Internet Use do not indicate authorship or whether it is an ad hoc instrument created along with the demographic information by the research team itself. Similarly, they are dichotomous questions that provide little value or are incomplete.

The results that are presented are either sparse or do not show the depth that might be expected at the beginning of the article. 

It does not present any major conclusions. 

Author Response

We appreciated your valuable comments. As discussed below, we have revised our manuscript with underlines based on your suggestions. Please let us know if we need to provide anything else regarding this revision.

Comment 1

The introduction is lax. The introduction does not provide any contextualisation that focuses or connects the difficulties that children with ADHD and their caregivers already have around behavioural symptoms and emotional symptoms.

Response

Thank you for your comment. We revised the Introduction section and added introduction for the difficulties that children with ADHD and their caregivers already have around behavioral and emotional symptoms before the COVID-19 pandemic as below.

The core symptoms of attention-deficit/hyperactivity disorder (ADHD) including inattention, hyperactivity and impulsivity increased the difficulties in learning among children with ADHD (CADHD) before the COVID-19 pandemic; CADHD encountered more academic problems such as lower grades, grade repetition, and increased school drop-out compared with those without ADHD [14–17].” Please refer to line 73-77

Studies have confirmed that not only ADHD symptoms but also emotional dysregulation and oppositional defiant disorder (ODD) have negative influences on CADHD’s learning [20,21]. Both emotional dysregulation and ODD were commonly presented in children with ADHD before the COVID-19 pandemic [22–24]. Children with ADHD were more likely than children without ADHD to experience an increase in inattentive, hyperactive/impulsive, oppositional/defiant and emotional symptoms during the COVID-19 pandemic [25].” Please refer to line 81-87.

Comment 2

The methodology is well presented, simple but appropriate to the data. However, some of the measurement instruments are not pre-scored. At a scientific level, it does not stand out too much, it presents obvious hypotheses, but it is necessary to make the results known given the exceptional conditions of the pandemic.
Response

Thank you for your comment. We added the introduction for the measurements (caregiver difficulties in managing child learning and worsened ADHD, ODD and emotional symptoms of children) as below. We also suggested that the reliability and validity of the measures warrants further study as below.

A previous study in 2020 using the same item found that caregiver difficulties in managing children’ learning at home was significantly associated with caregivers’ poor mental health state in the COVID-19 pandemic [32].” Please refer to line 158-160.

A previous study in 2020 using the same items found that children’s worsened ADHD, ODD and emotional symptoms were significantly associated with caregivers’ poor mental health state in the COVID-19 pandemic [32].” Please refer to line 166-168.

“…although the previous study using the same research questions found that caregivers’ difficulty in managing child’s learning at home and children’s worsened ADHD, ODD and emotional symptoms were significantly associated with caregivers’ poor mental health state in the COVID-19 pandemic [32], the validity and reliability of the questions assessing caregivers’ difficulties in managing child’s learning and children’s worsened ADHD, ODD and emotional symptoms warrant further study.” Please refer to line 369-374.

Comment 3

The Participants section should note or name the presentation of the sample in more detail in Table 1. 

Response

We expanded the description for the characteristics of the participants as below. Please refer to line 224-234.

Table 1 additionally reports the children’s demographics, treatment, ADHD and ODD symptoms, parenting styles information, and caregivers’ difficulties in managing child’s learning. The children had a mean age of 9.61 (SD = 2.39) years. Approximately four-fifths of the children (n = 200; 79.4%) were boys, and the majority of them received regular medical treatment (n = 212; 84.1%). Children’s mean (SD) scores for the dimensions of inattention, hyperactivity/impulsivity, and ODD symptoms were 12.88 (5.83), 9.93 (6.17), and 9.33 (5.92), respectively. The mean (SD) PBI scores for the affection care, overprotection, and authoritarian parenting dimensions were 37.08 (5.16), 13.75 (3.32), and 12.24 (2.67), respectively. Over four-fifths of the caregivers had difficulty in asking their children to learn at home (n = 215; 85.3%); more than one-quarter had difficulty in taking their children to afterschool learning programs (n = 71; 28.2%).

Comment 4

The instruments Caregiver Difficulties in Managing Children's Learning and Changes in ADHD, ODD, and Emotional Symptoms and Internet Use do not indicate authorship or whether it is an ad hoc instrument created along with the demographic information by the research team itself. Similarly, they are dichotomous questions that provide little value or are incomplete.

Response

As described in Response to Comment 2, we added the introduction for the measurements (caregiver difficulties in managing child learning and worsened ADHD, ODD and emotional symptoms of children). We also suggested that the reliability and validity of the measures warrants further study.

Comment 5

The results that are presented are either sparse or do not show the depth that might be expected at the beginning of the article. 

Response

We rewrote the results of multivariate logistic regression models as below to connect them with the aims and hypotheses of this study. Please refer to line 249-262.

The results of multivariate logistic regression models are shown in Table 3. Worsened anger (AOR = 4.36; 95% CI = 1.56, 12.18) was significantly associated with higher caregiver difficulty in asking children to learn at home, whereas parental overprotection was significantly associated with lower caregiver difficulties in asking children to learn at home (AOR = 0.88; 95% CI = 0.78, 0.99). Worsened attention, hyperactivity, impulsivity, opposition, revenge, depression, anxiety and Internet use, and parenting styles of affection/care and authoritarianism were not significantly associated with caregiver difficulties in asking children learning at home.

Worsened hyperactivity (AOR = 3.46; 95% CI = 1.73, 6.91) and opposition (AOR = 1.99; 95% CI = 1.01, 3.95) were significantly associated with higher caregiver difficulty in taking children to attend afterschool learning programs. Worsened attention, impulsivity, anger, revenge, depression, anxiety and Internet use, and parenting styles were not significantly associated with caregiver difficulties in taking children to attend afterschool learning programs.

Comment 6

It does not present any major conclusions. 

Response

We rewrote the conclusion section to include the major conclusions based on the results of this study. Please refer to line 380-394.

The present study found a high proportion of caregivers had difficulties in managing the learning of their CADHD at home and attending afterschool learning programs during the COVID-19 pandemic. Because that caregivers have a key role in providing support for their children’s home learning during the COVID-19 pandemic, enhancing caregivers’ managing skills to support children’s learning motivation and ability during the COVID-19 pandemic is necessary to reduce the difficulties of caregivers. The after-effects of learning interference from the COVID-19 pandemic among children with ADHD warrant investigation through follow-up studies. Children’s worsened anger, hyperactivity and opposition were significantly associated with higher caregiver difficulty in managing their CADHD’s learning; therefore, children’s ADHD and ODD symptoms should also be routinely assessed and intervened. Caregivers with higher parental overprotection reported lower difficulty in asking children to learn at home; however, the long-term influence of parenting overprotection on the learning of CADHD warrants further study. Behavioral and medication interventions should be conducted to maintain the psychological well-being of CADHD during the COVID-19 pandemic.

Round 2

Reviewer 1 Report

The authors efforts at revision have enhanced an already very good manuscript. The additional clarity and discussion are much appreciated. The authors provided more clarity on the interrelatedness of CADHD and ODD and why both figured prominently in the analyses. Exploring the top and bottom 25% of worsened ADHD and ODD makes the results substantially more important for policy and practice.